# Your GAN is Secretly an Energy-based Model and You Should Use Discriminator Driven Latent Sampling

**Tong Che**[*][1][2]    **Ruixiang Zhang**[*][1][2]    **Jascha Sohl-Dickstein**[3]

**Hugo Larochelle**[1][3]    **Liam Paull**[1][2]    **Yuan Cao**[3]    **Yoshua Bengio**[1][2]

[1]Mila    [2] Université de Montréal    [3]Google Brain
{tong.che, ruixiang.zhang}@umontreal.ca

## Abstract

The sum of the implicit generator log-density $\log p_g$ of a GAN with the logit score of the discriminator defines an energy function which yields the true data density when the generator is imperfect but the discriminator is optimal. This makes it possible to improve on the typical generator (with implicit density $p_g$). We show that samples can be generated from this modified density by sampling in *latent space* according to an energy-based model induced by the sum of the latent prior log-density and the discriminator output score. We call this process of running Markov Chain Monte Carlo in the latent space, and then applying the generator function, Discriminator Driven Latent Sampling (DDLS). We show that DDLS is highly efficient compared to previous methods which work in the high-dimensional pixel space, and can be applied to improve on previously trained GANs of many types. We evaluate DDLS on both synthetic and real-world datasets qualitatively and quantitatively. On CIFAR-10, DDLS substantially improves the Inception Score of an off-the-shelf pre-trained SN-GAN [1] from $8.22$ to $9.09$ which is comparable to the class-conditional BigGAN [2] model. This achieves a new state-of-the-art in the unconditional image synthesis setting without introducing extra parameters or additional training.

## 1   Introduction

Generative Adversarial Networks (GANs) [3] are state-of-the-art models for a large variety of tasks such as image generation [4], semi-supervised learning [5], image editing [6], image translation [7], and imitation learning [8]. The GAN framework consists of two neural networks, the generator $G$ and the discriminator $D$. The optimization process is formulated as an adversarial game, with the generator trying to fool the discriminator and the discriminator trying to better classify samples as real or fake.

Despite the ability of GANs to generate high-resolution, sharp samples, the samples of GAN models sometimes contain bad artifacts or are even not recognizable [9]. It is conjectured that this is due to the inherent difficulty of generating high dimensional complex data, such as natural images, and the optimization challenge of the adversarial formulation. In order to improve sample quality, conventional sampling techniques, such as increasing the temperature, are commonly adopted for GAN models [2]. Recently, new sampling methods such as Discriminator Rejection Sampling (DRS) [10], Metropolis-Hastings Generative Adversarial Network (MH-GAN) [11], and Discriminator Optimal

---

[*]Equal contribution. Ordering determined by coin flip.

Transport (DOT) [12] have shown promising results by utilizing the information provided by both the generator and the discriminator. However, these sampling techniques are either inefficient or lack theoretical guarantees, possibly reducing the sample diversity and making the mode dropping problem more severe.

In this paper, we show that GANs can be better understood through the lens of Energy-Based Models (EBM). In our formulation, GAN generators and discriminators collaboratively learn an "implicit" energy-based model. However, efficient sampling from this energy based model directly in pixel space is *extremely* challenging for several reasons. One is that there is no tractable closed form for the implicit energy function in pixel space. This motivates an intriguing possibility: that Markov Chain Monte Carlo (MCMC) sampling may prove more tractable in the GAN's latent space.

Surprisingly, we find that the implicit energy based model defined jointly by a GAN generator and discriminator takes on a simpler, tractable form when it is written as an energy-based model over the generator's latent space. In this way, we propose a theoretically grounded way of generating high quality samples from GANs through what we call Discriminator Driven Latent Sampling (DDLS). DDLS leverages the information contained in the discriminator to re-weight and correct the biases and errors in the generator. Through experiments, we show that our proposed method is highly efficient in terms of mixing time, is generally applicable to a variety of GAN models (e.g. Minimax, Non-Saturating, and Wasserstein GANs), and is robust across a wide range of hyper-parameters. An energy-based model similar to our work is also obtained simultaneously in independent work [13] in the form of an approximate MLE lower bound.

We highlight our main contributions as follows:

- We provide more evidence that it is beneficial to sample from the energy-based model defined both by the generator and the discriminator instead of from the generator only.

- We derive an equivalent formulation of the pixel-space energy-based model in the latent space, and show that sampling is much more efficient in the latent space.

- We show experimentally that samples from this energy-based model are of higher quality than samples from the generator alone.

- We show that our method can approximately extend to other GAN formulations, such as Wasserstein GANs.

## 2 Background

### 2.1 Generative Adversarial Networks

GANs [3] are a powerful class of generative models defined through an adversarial minimax game between a generator network $G$ and a discriminator network $D$. The generator $G$ takes a latent code $z$ from a prior distribution $p(z)$ and produces a sample $G(z) \in X$. The discriminator takes a sample $x \in X$ as input and aims to classify real data from fake samples produced by the generator, while the generator is asked to fool the discriminator as much as possible. We use $p_d$ to denote the true data-generating distribution and $p_g$ to denote the implicit distribution induced by the prior and the generator network. The standard non-saturating training objective for the generator and discriminator is defined as:

$$
\begin{aligned}
L_D &= -\mathbb{E}_{x \sim p_{\text{data}}}[\log D(x)] - \mathbb{E}_{z \sim p_z}[\log(1 - D(G(z)))] \\
L_G &= -\mathbb{E}_{z \sim p_z}[\log D(G(z))]
\end{aligned}
\tag{1}
$$

Wassertein GANs (WGAN) [14] are a special family of GAN models. Instead of targeting a Jensen-Shannon distance, they target the 1-Wasserstein distance $W(p_g, p_d)$. The WGAN discriminator objective function is constructed using the Kantorovich duality $\max_{D \in \mathcal{L}} \mathbb{E}_{p_{\text{data}}}[D(x)] - \mathbb{E}_{p_g}[D(x)]$ where $\mathcal{L}$ is the set of 1-Lipstchitz functions.

### 2.2 Energy-Based Models and Langevin Dynamics

An energy-based model (EBM) is defined by a Boltzmann distribution $p(x) = e^{-E(x)}/Z$, where $x \in \mathcal{X}$, $\mathcal{X}$ is the state space, and $E(x) : \mathcal{X} \to \mathbb{R}$ is the energy function. Samples are typically

generated from $p(x)$ by an MCMC algorithm. One common MCMC algorithm in continuous state spaces is Langevin dynamics, with update equation $x_{i+1} = x_i - \frac{\epsilon}{2}\nabla_x E(x) + \sqrt{\epsilon}n, n \sim N(0, I)$[2].

One solution to the problem of slow-sampling Markov Chains is to perform sampling using a carfefully crafted latent space [15, 16]. Our method shows how one can execute such latent space MCMC in GAN models.

## 3 Methodology

### 3.1 GANs as an Energy-Based Model

Suppose we have a GAN model trained on a data distribution $p_d$ with a generator $G(z)$ with generator distribution $p_g$ and a discriminator $D(x)$. We assume that $p_g$ and $p_d$ have the same support. This can be guaranteed by adding small Gaussian noise to these two distributions.

The training of GANs is an adversarial game which generally does not converge to the optimal generator, so usually $p_d$ and $p_g$ do not match perfectly at the end of training. However, the discriminator provides a quantitative estimate for how much these two distributions (mis)match. Let's assume the discriminator is near optimality, namely [3] $D(x) \approx \frac{p_d(x)}{p_d(x)+p_g(x)}$. From this equation, let $d(x)$ be the logit of $D(x)$, in which case $\frac{p_d(x)}{p_d(x)+p_g(x)} = \frac{1}{1+\frac{p_g(x)}{p_d(x)}} \approx \frac{1}{1+\exp(-d(x))}$, and we have $e^{d(x)} \approx p_d/p_g$, and $p_d(x) \approx p_g(x)e^{d(x)}$. Normalization of $p_g(x)e^{d(x)}$ is not guaranteed, and it will not typically be a valid probabilistic model. We therefore consider the energy-based model $p_d^* = p_g(x)e^{d(x)}/Z_0$, where $Z_0$ is a normalization constant. Intuitively, this formulation has two desirable properties. First, as we elaborate later, if $D = D^*$ where $D^*$ is the optimal discriminator, then $p_d^* = p_d$. Secondly, it corrects the bias in the generator via weighting and normalization. If we can sample from this distribution, it should improve our samples.

There are two difficulties in sampling efficiently from $p_d^*$:

1. Doing MCMC in pixel space to sample from the model is impractical due to the high dimensionality and long mixing time.

2. $p_g(x)$ is implicitly defined and its density cannot be computed directly.

In the next section we show how to overcome these two difficulties.

### 3.2 Rejection Sampling and MCMC in Latent Space

Our approach to the above two problems is to formulate an equivalent energy-based model in the latent space. To derive this formulation, we first review rejection sampling [17]. With $p_g$ as the proposal distribution, we have $e^{d(x)}/Z_0 = p_d^*(x)/p_g(x)$. Denote $M = \max_x p_d^*(x)/p_g(x)$ (this is well-defined if we add a Gaussian noise to the output of the generator and $x$ is in a compact space). If we accept samples from proposal distribution $p_g$ with probability $p_d^*/(Mp_g)$, then the samples we produce have the distribution $p_d^*$.

We can alternatively interpret the rejection sampling procedure above as occurring in the latent space $z$. In this interpretation, we first sample $z$ from $p(z)$, and then perform rejection sampling on $z$ with acceptance probability $e^{d(G(z))}/(MZ_0)$. Only once a latent sample $z$ has been accepted do we generate the pixel level sample $x = G(z)$.

This rejection sampling procedure on $z$ induces a new probability distribution $p_t(z)$. To explicitly compute this distribution we need to conceptually reverse the definition of rejection sampling. We formally write down the "reverse" lemma of rejection sampling as Lemma 1, to be used in our main theorem.

**Lemma 1.** *On space $X$ there is a probability distribution $p(x)$. $r(x) : X \to [0, 1]$ is a measurable function on $X$. We consider sampling from $p$, accepting with probability $r(x)$, and repeating this*

*procedure until a sample is accepted. We denote the resulting probability measure of the accepted samples $q(x)$. Then we have $q(x) = p(x)r(x) / Z$, where $Z = \mathbb{E}_p[r(x)]$.*

Namely, we have the prior proposal distribution $p_0(z)$ and an acceptance probability $r(z) = e^{d(G(z))} / (MZ_0)$. We want to compute the distribution after the rejection sampling procedure with $r(z)$. With Lemma 1, we can see that $p_t(z) = p_0(z)r(z) / Z'$. We expand on the details in our main theorem.

### 3.3 Main Theorem

**Theorem 1.** *Assume $p_d$ is the data generating distribution, and $p_g$ is the generator distribution induced by the generator $G : \mathcal{Z} \to \mathcal{X}$, where $\mathcal{Z}$ is the latent space with prior distribution $p_0(z)$. Define Boltzmann distribution $p_d^* = e^{\log p_g(x) + d(x)} / Z_0$, where $Z_0$ is the normalization constant.*

*Assume $p_g$ and $p_d$ have the same support. We address the case when this assumption does not hold in Corollary 2. Further, let $D(x)$ be the discriminator, and $d(x)$ be the logit of $D$, namely $D(x) = \sigma(d(x))$. We define the energy function $E(z) = -\log p_0(z) - d(G(z))$, and its Boltzmann distribution $p_t(z) = e^{-E(z)} / Z$. Then we have:*

1. *$p_d^* = p_d$ when $D$ is the optimal discriminator.*
2. *If we sample $z \sim p_t$, and $x = G(z)$, then we have $x \sim p_d^*$. Namely, the induced probability measure $G \circ p_t = p_d^*$.*

*Proof.* Please see Appendix A. $\qquad\qquad\qquad\qquad\qquad\qquad\qquad\qquad\qquad\qquad\square$

Interestingly, $p_t(z)$ has the form of an energy-based model, $p_t(z) = e^{-E(z)} / Z'$, with *tractable* energy function $E(z) = -\log p_0(z) - d(G(z))$. In order to sample from this Boltzmann distribution, one can use an MCMC sampler, such as Langevin dynamics or Hamiltonian Monte Carlo. We defer the proofs and the MCMC algorithm to our Supplemental Material.

### 3.4 Sampling Wasserstein GANs with Langevin Dynamics

Wasserstein GANs are different from original GANs in that they target the Wassertein loss. Although when the discriminator is trained to optimality, the discriminator can recover the Kantorovich dual [14] of the optimal transport between $p_g$ and $p_d$, the target distribution $p_d$ cannot be exactly recovered using the information in $p_g$ and $D$[3]. However, in the following we show that in practice, the optimization of WGAN can be viewed as an approximation of an energy-based model, which can also be sampled with our method.

The objectives of Wasserstein GANs can be summarized as:

$$L_D = \mathbb{E}_{p_g}[D(x)] - \mathbb{E}_{p_d}[D(x)] \quad , \quad L_G = -\mathbb{E}_{p_0}[D(G(z))] \qquad (2)$$

where $D$ is restricted to be a $K$-Lipschitz function.

On the other hand, consider a new energy-based generative model (which also has a generator and a discriminator) trained with the following objectives (for detailed algorithm, please refer to our Supplemental Material):

1. Discriminator training phase (D-phase). Unlike GANs, our energy-based model tries to match the distribution $p_t(x) = p_g(x)e^{D_\phi(x)} / Z$ with the data distribution $p_d$, where $p_t(x)$ can be interpreted as an EBM with energy $D_\phi(x) - \log p_g(x)$. In this phase, the generator is kept fixed, and the discriminator is trained.
2. Generator training phase (G-phase). The generator is trained such that $p_g(x)$ matches $p_t(x)$, in this phase we treat $D$ as fixed and train $G$.

In the D-phase, we are training an EBM with data from $p_d$. The gradient of the KL-divergence (which is our loss function for D-phase) can be written as [18]:

$$\nabla_\phi \text{KL}(p_d || p_t) = \mathbb{E}_{p_t}[\nabla_\phi D(x)] - \mathbb{E}_{p_d}[\nabla_\phi D(x)] \qquad (3)$$

Namely we are trying to maximize $D$ on real data and trying to minimize it on fake data. Note that the fake data distribution $p_t$ is a function of both the generator and discriminator, and cannot be sampled directly. As with other energy-based models, we can use an MCMC procedure such as Langevin dynamics to generate samples from $p_t$ [19].

In the G-phase, we can train the model with the gradient of KL-divergence $\mathrm{KL}(p_g \, || \, p_t')$ as our loss. Let $p_t'$ be a fixed copy of $p_t$, we can compute the gradient as (see the Appendix for more details):

$$\nabla_\theta \mathrm{KL}(p_g \, || \, p_t') = -\mathbb{E}[\nabla_\theta D(G(z))]. \tag{4}$$

Note that the losses above coincide with what we are optimizing in WGANs, with two differences:

1. In WGAN, we optimize $D$ on $p_g$ instead of $p_t$. This may not be a big difference in practice, since as training progresses $p_t$ is expected to approach $p_g$, as the optimizing loss for the generator explicitly acts to bring $p_g$ closer to $p_t$ (Equation 4). Moreover, it has recently been found in LOGAN [20] that optimizing $D$ on $p_t$ rather than $p_g$ can lead to better performance.

2. In WGAN, we impose a Lipschitz constraint on $D$. This constraint can be viewed as a smoothness regularizer. Intuitively it will make the distribution $p_t(x) = p_g(x)e^{-D_\phi(x)}/Z$ more "flat" than $p_d$, but $p_t(x)$ (which lies in a distribution family parameterized by $D$) remains an approximator to $p_d$ subject to this constraint.

Thus, we can conclude that for a Wasserstein GAN with discriminator $D$, WGAN approximately optimizes the KL divergence of $p_t = p_g(x)e^{-D(x)}/Z$ with $p_d$, with the constraint that $D$ is $K$-Lipschitz. This suggests that one can also perform DDLS on the WGAN latent space to generate improved samples, using an energy function $E(z) = -\log p_0(z) - D(G(z))$.

### 3.5 Practical Issues and the Mode Dropping Problem

Mode dropping is a major problem in training GANs. In our main theorem it is assumed that $p_g$ and $p_d$ have the same support. We also assumed that $G : \mathcal{Z} \to \mathcal{X}$ is a deterministic function. Thus, if $G$ cannot recover some of the modes in $p_d$, $p_d^*$ also cannot recover these modes.

However, we can partially solve the mode dropping problem by introducing an additional Gaussian noise $z' \sim N(0, 1; z') = p_1(z')$ to the output of the generator, namely we define the new deterministic generator $G^*(z, z') = G(z) + \epsilon z'$. We treat $z'$ as a part of the generator, and do DDLS on joint latent variables $(z, z')$. The Langevin dynamics on this joint energy will help the model to move data points that are a little bit off-mode to the data manifold, and we have the follwing Corollary:

**Corollary 1.** *Assume $p_d$ is the data generating distribution with small Gaussian noise added. The generator $G : \mathcal{Z} \to \mathcal{X}$ is a deterministic function, where $\mathcal{Z}$ is the latent space endowed with prior distribution $p_0(z)$. Assume $z' \sim p_1(z') = N(0, 1; z)$ is an additional Gaussian noise variable with $\dim z' = \dim \mathcal{X}$. Let $\epsilon > 0$, denote the distribution of the extended generator $G^*(z, z') = G(z) + \epsilon z'$ as $p_g$. $D(x)$ is the discriminator trained between $p_g$ and $p_d$. Let $d(x)$ be the logit of $D$, namely $D(x) = \sigma(d(x))$. Define $p_d^* = e^{\log p_g(x) + d(x)}/Z_0$, where $Z_0$ is the normalization constant. We define the energy function in the extended latent space $E(z, z') = -\log p_0(z) - \log p_1(z') - d(G^*(z, z'))$, and its Boltzmann distribution $p_t(z, z') = e^{-E(z, z')}/Z$. Then we have:*

1. *$p_d^* = p_d$ when $D$ is the optimal discriminator.*
2. *If we sample $(z, z') \sim p_t$, and $x = G^*(z, z')$, then we have $x \sim p_d^*$. Namely, the induced probability measure $G^* \circ p_t = p_d^*$.*

*Proof.* Please see Appendix A. □

## 4 Related Work

Previous work has considered utilizing the discriminator to achieve better sampling for GANs. Discriminator rejection sampling [10] and Metropolis-Hastings GANs [11] use $p_g$ as the proposal distribution and $D$ as the criterion of acceptance or rejection. However, these methods are inefficient as they may need to reject a wechlot of samples. Intuitively, one major drawback of these methods is that since they operate in the pixel space, their algorithm can use discriminators to reject samples

when they are bad, but cannot easily guide latent space updates using the discriminator which would improve these samples. The advantage of DDLS over DRS or MH-GAN is similar to the advantage of SGD over zero-th order optimization algorithms.

Trained classifiers have similarly been used to correct probabilistic models in other contexts [21]. Discriminator optimal transport (DOT) [12] is another way of sampling GANs. They use deterministic gradient descent in the latent space to get samples with higher $D$-values, However, since $p_g$ and $D$ cannot recover the data distribution exactly, DOT has to make the optimization local in a small neighborhood of generated samples, which hurts the sample performance. Also, DOT is not guaranteed to converge to the data distribution even under ideal assumptions ($D$ is optimal).

Other previous work considered the usage of probabilistic models defined jointly by the generator and discriminator. In [22], the authors use the idea of training an EBM defined jointly by a generator and an additional critic function in the text generation setting. [23] uses an additional discriminator as a bias corrector for generative models via importance weighting. [24] considered rejection sampling in latent space in encoder-decoder models.

Energy-based models [25–33] have gained significant attention in recent years. Most work focuses on the maximum likelihood learning of energy-based models [34–36]. Other work has built new connections between energy based models and classifiers [37]. The primary difficulty in training energy-based models comes from effectively estimating and sampling the partition function. The contribution to training from the partition function can be estimated via MCMC [35, 38, 39], via training another generator network [40, 41], or via surrogate objectives to maximum likelihood [42–44]. The connection between GANs and EBMs has been studied by many authors [45–48]. Our paper can be viewed as establishing a new connection between GANs and EBMs which allows efficient latent MCMC sampling.

## 5 Experimental results

In this section we present a set of experiments demonstrating the effectiveness of our method on both synthetic and real-world datasets. In section 5.1 we illustrate how the proposed method, DDLS, can improve the distribution modeling of a trained GAN and compare with other baseline methods. In section 5.2 we show that DDLS can improve the sample quality on real world datasets, both qualitatively and quantitatively.

### 5.1 Synthetic dataset

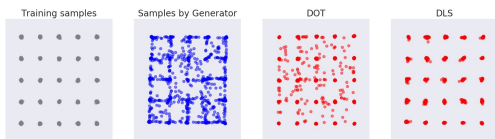 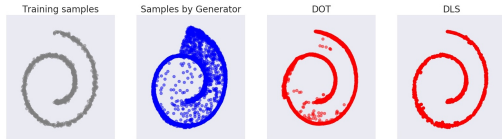

Figure 1: DDLS, generator alone, and generator + DOT, on 2d mixture of Gaussians distribution

Figure 2: DDLS, the generator alone, and generator + DOT, on the swiss roll dataset.

Table 1: DDLS suffers less from mode dropping when modeling the 2d synthetic distribution in Figure 1. Table shows number of recovered modes, and fraction of "high quality" (less than four standard deviations from mode center) recovered modes.

|  | # recovered modes | % "high quality" | std "high quality" |
|---|---|---|---|
| Generator only | $24.8 \pm 0.2$ | $70 \pm 9$ | $0.11 \pm 0.01$ |
| DRS | $24.8 \pm 0.2$ | $90 \pm 2$ | $\mathbf{0.10 \pm 0.01}$ |
| GAN w. DDLS | $24.8 \pm 0.2$ | $\mathbf{98 \pm 2}$ | $\mathbf{0.10 \pm 0.01}$ |

Following the same setting used in [10–12], we apply DDLS to a WGAN model trained on two synthetic datasets, 25-gaussians and Swiss Roll, and investigate the effect and performance of the proposed sampling method.

Table 2: DDLS has lower Earth Mover's Distance (EMD) to the true distribution for the 2d synthetic distribution in Figure 1, and matches the performance of DOT on the Swiss Roll distribution.

|  | EMD 25-Gaussian | EMD Swiss Roll |
|---|---|---|
| Generator only([12]) | 0.052(08) | 0.021(05) |
| DOT([12]) | 0.052(10) | **0.020**(06) |
| Generator only(Our imple.) | 0.043(04) | 0.026(03) |
| GAN as EBM with DDLS | **0.036**(04) | **0.020**(05) |

Table 3: Inception and FID scores on CIFAR-10 and CelebA (grouped by corresponding baseline modles), showing a substantial quantitative advantage from DDLS, compared to MH-GAN [11], DRS [10] and DOT [12] using the same architecture.

| Model | CIFAR-10 | | CelebA |
|---|---|---|---|
|  | Inception | FID | Inception |
| DCGAN w/o DRS or MH-GAN | 2.8789 | - | 2.3317 |
| DCGAN w/ DRS(cal) [10] | 3.073 | - | 2.869 |
| DCGAN w/ MH-GAN(cal) [11] | 3.379 | - | 3.106 |
| WGAN w/o DRS or MH-GAN | 3.0734 | - | 2.7876 |
| WGAN w/ DRS(cal) [10] | 3.137 | - | 2.861 |
| WGAN w/ MH-GAN(cal) [11] | 3.305 | - | 2.889 |
| **Ours: DCGAN w/ DDLS** | **3.681** | - | **3.372** |
| **Ours: WGAN w/ DDLS** | **3.614** | - | **3.093** |
| PixelCNN [49] | 4.60 | 65.93 | - |
| EBM [35] | 6.02 | 40.58 | - |
| WGAN-GP [50] | $7.86 \pm .07$ | 36.4 | - |
| ProgressiveGAN [51] | $8.80 \pm .05$ | - | - |
| NCSN [52] | $8.87 \pm .12$ | 25.32 | - |
| ResNet-SAGAN w/o DOT | $7.85 \pm .11$ | 21.53 | - |
| ResNet-SAGAN w/ DOT | $8.50 \pm .12$ | 19.71 | - |
| SNGAN w/o DDLS | $8.22 \pm .05$ | 21.7 | - |
| Ours: SNGAN w/ DDLS | $9.05 \pm .11$ | 15.76 | - |
| **Ours: SNGAN w/ DDLS(cal)** | $\mathbf{9.09 \pm 0.10}$ | **15.42** | - |

**Implementation details** We follow the same synthetic experiments design as in DOT [12], while parameterizing the prior with a standard normal distribution instead of a uniform distribution. Please refer to C.1 for more details.

**Qualitative results** With the trained generator and discriminator, we generate 5000 samples from the generator, then apply DDLS in latent space to obtain enhanced samples. We also apply the DOT method as a baseline. All results are depicted in Figure 1 and Figure 2 together with the target dataset samples. For the 25-Gaussian dataset we can see that DDLS recovered and preserved all modes while significantly eliminating spurious modes compared to a vanilla generator and DOT. For the Swiss Roll dataset we can also observe that DDLS successfully improved the distribution and recovered the underlying low-dimensional manifold of the data distribution. This qualitative evidence supports the hypothesis that our GANs as energy based model formulation outperforms the noisy implicit distribution induced by the generator alone.

**Quantitative results** We first examine the performance of DDLS quantitavely by using the metrics proposed by DRS [10]. We generate $10,000$ samples with the DDLS algorithm, and each sample is assigned to its closest mixture component. A sample is of "high quality" if it is within four standard deviations of its assigned mixture component, and a mode is successfully "recovered" if at least one high-quality sample is assigned to it.

As shown in Table 1, our proposed model achieves a higher "high-quality" ratio. We also investigate the distance between the distribution induced by our GAN as EBM formulation and the true data distribution. We use the Earth Mover's Distance (EMD) between the two corresponding empirical distributions as a surrogate, as proposed in DOT [12]. As shown in Table 2, the EMD between our sampling distribution and the ground-truth distribution is significantly below the baselines. Note that we use our own re-implementation, and numbers differ slightly from those previously published.

## 5.2 CIFAR-10 and CelebA

In this section we evaluate the performance of the proposed DDLS method on the CIFAR-10 dataset and CelebA dataset.

**Implementation details** We provide detailed description of baseline models, DDLS hyper-parameters and evaluation protocol in C.2.

**Quantitative results** We evaluate the quality and diversity of generated samples via the Inception Score [53] and Fréchet Inception Distance (FID) [54]. In Table 3, we show the Inception score improvements from DDLS on CIFAR-10 and CelebA, compared to MH-GAN [11] and DRS [10], following the same evaluation protocol and using the same baseline models (DCGAN and WGAN) in [11]. On CIFAR-10, we applied DDLS to the unconditional generator of SN-GAN to generate 50000 samples and report all results in Table 3. We found that the proposed method significantly improves the Inception Score of the baseline SN-GAN model from 8.22 to 9.09 and reduces the FID from 21.7 to 15.42. Our unconditional model outperforms previous state-of-the-art GANs and other sampling-enhanced GANs [10–12] and even approaches the performance of conditional BigGANs [2] which achieves an Inception Score 9.22 and an FID of 14.73, *without the need of additional class information, training and parameters*.

**Qualitative results** We illustrate the process of Langevin dynamics sampling in latent space in Figure 3 by generating samples for every 10 iterations. We find that our method helps correct the errors in the original generated image, and makes changes towards more semantically meaningful and sharp output by leveraging the pre-trained discriminator. We include more generated samples for visualizing the Langevin dynamics in the appendix. To demonstrate that our model is not simply memorizing the CIFAR-10 dataset, we find the nearest neighbors of generated samples in the training dataset and show the results in Figure 4.

**Mixing time evaluation** MCMC sampling methods often suffer from extremely long mixing times, especially for high-dimensional multi-modal data. For example, more than 600 MCMC iterations are need to obtain the most performance gain in MH-GAN [11] on real data. We demonstrate the sampling efficiency of our method by showing that we can expect a much shorter time to achieve competitive performance by migrating the Langevin sampling process to the latent space, compared to sampling in high-dimensional multi-modal pixel space. We evaluate the Inception Score and the energy function for every 10 iterations of Langevin dynamics and depict the results in Figure 5 in Appendix.

## 5.3 ImageNet

Table 4: Inception score for ImageNet, showing the quantitative advantage of DDLS.

| Model | Inception |
|---|---|
| SNGAN [1] | 36.8 |
| cGAN w/o DOT | 36.23 |
| cGAN w/ DOT | 37.29 |
| Ours: SNGAN w/ DDLS | 40.2 |

In this section we evaluate the performance of the proposed DDLS method on the ImageNet dataset.

**Implementation details** As with CIFAR-10, we adopt the Spectral Normalization GAN (SN-GAN) [1] as our baseline GAN model. We take the publicly available pre-trained models of SN-GAN and apply DDLS. Fine tuning is performed on the discriminator, as described in Section 5.2. Implementation choices are otherwise the same as for CIFAR-10, with additional details in the appendix. We show the quantitative results in Table 4, where we substantially outperform the baseline.

## 6 Conclusion and Future Work

In this paper, we have shown that a GAN's discriminator can enable better modeling of the data distribution with Discriminator Driven Latent Sampling (DDLS). The intuition behind our model is that learning a generative model to do structured generative prediction is usually more difficult than learning a classifier, so the errors made by the generator can be significantly corrected by the

discriminator. The major advantage of DDLS is that it allows MCMC sampling in the latent space, which enables efficient sampling and better mixing.

For future work, we are exploring the inclusion of additional Gaussian noise variables in each layer of the generator, treated as latent variables, such that DDLS can be used to provide a correcting signal for each layer of the generator. We believe that this will lead to further sampling improvements. Also, prior work on VAEs has shown that learned re-sampling or energy-based transformation of their priors can be effective [55, 56]. It would thus be particularly interesting to explore whether VAE-based models can be improved by constructing an energy based model for their prior based on an auxiliary discriminator.

## 7 Broader Impact

The development of powerful generative models which can generate fake images, audio, and video which appears realistic is a source of acute concern [57], and can enable fake news or propaganda. On the other hand these same technologies also enable assistive technologies like text to speech [58], new art forms [59, 60], and new design technologies [61].

Our work enables the creation of more powerful generative models. As such it is multi-use, and may result in both positive and negative societal consequences. However, we believe that improving scientific understanding tends also to improve the human condition [62] – so in the absence of a reason to expect specific harm, we believe that in expectation our work will have a positive impact on the world.

One potential benefit of our work is that it leads to better calibrated GANs, which are less likely to simply drop under-represented sample classes from their generated output. The tendency of machine learning models to produce worse outcomes for groups which are underrepresented in their training data (in terms of race, gender, or otherwise) is well documented [63]. The use of our technique should produce generative models which are slightly less prone to this type of bias.

## Acknowledgments and Disclosure of Funding

The authors are grateful to Ben Poole and the anonymous reviewers for proof-reading the paper and suggesting improvements. This work was supported by CIFAR and Compute Canada.

## Footnotes

[2]Langevin dynamics are guaranteed to exactly sample from the target distribution $p(x)$ as $\epsilon \to 0, i \to \infty$. In practice we will use a small, finite, value for $\epsilon$ in our experiments. In this case, one can add an additional layer of M-H sampling, resulting in the MALA algorithm, to eliminate biases.

[3]In Tanaka [12], the authors claim that it is possible to recover $p_d$ with $D$ and $p_g$ in WGAN in certain metrics, but we show in the Appendix that their assumptions don't hold and in the $L^1$ metric, which WGAN uses, it is not possible to recover $p_d$.

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
