[Supplementary Material]

# A  Proofs

## A.1  Proof of the Main Theorem

**Lemma 2.** *On space $X$ there is a probability distribution $p(x)$. $r(x) : X \to [0,1]$ is a measurable function on $X$. We consider sampling from $p$, accepting with probability $r(x)$, and repeating this procedure until a sample is accepted. We denote the resulting probability measure of the accepted samples $q(x)$. Then we have:*

$$q(x) = p(x)r(x) \,/\, Z, \; Z = \mathbb{E}_p[r(x)]. \tag{5}$$

*Proof.* From the definition of rejection sampling, we can see that in order to get the distribution $q(x)$, we can sample $x$ from $p(x)$ and do rejection sampling with probability $r'(x) = q(x) \,/\, (Mp(x))$, where $M \geq q(x) \,/\, p(x)$ for all $x$. So we have $r'(x) = r(x) \,/\, (ZM)$. If we choose $M = 1 \,/\, Z$, then from $r(x) \leq 1$ for all $x$, we can see that $M$ satisfies $M \geq q(x) \,/\, p(x) = r(x) \,/\, Z$, for all $x$. So we can choose $M = 1 \,/\, Z$, resulting in $r(x) = r'(x)$. $\qquad\square$

**Theorem 2.** *Assume $p_d$ is the data generating distribution, and $p_g$ is the generator distribution induced by the generator $G : \mathcal{Z} \to \mathcal{X}$, where $\mathcal{Z}$ is the latent space with prior distribution $p_0(z)$. Define $p_d^* = e^{\log p_g(x) + d(x)} \,/\, Z_0$, where $Z_0$ is the normalization constant.*

*Assume $p_g$ and $p_d$ have the same support. This assumption is typically satisfied when $\dim(z) \geq \dim(x)$. We address the case that $\dim(z) < \dim(x)$ in Corollary 2. Further, let $D(x)$ be the discriminator, and $d(x)$ be the logit of $D$, namely $D(x) = \sigma(d(x))$. We define the energy function $E(z) = -\log p_0(z) - d(G(z))$, and its Boltzmann distribution $p_t(z) = e^{-E(z)} \,/\, Z$. Then we have:*

1. *$p_d^* = p_d$ when $D$ is the optimal discriminator.*

2. *If we sample $z \sim p_t$, and $x = G(z)$, then we have $x \sim p_d^*$. Namely, the induced probability measure $G \circ p_t = p_d^*$.*

*Proof.* (1) follows from the fact that when $D$ is optimal, $D(x) = \frac{p_g}{p_d + p_g}$, so $D(x) = \sigma(\log p_d - \log p_g)$, which implies that $d(x) = \log p_d - \log p_g$ (which is finite on the support of $p_g$ due to the fact that they have the same support). Thus, $p_d^*(x) = p_d(x) \,/\, Z_0$, we must have $Z_0 = 1$ for normalization, so $p_d^* = p_d$.

For (2), for samples $x \sim p_g$, if we do rejection sampling with probability $p_d^*(x) \,/\, (Mp_g(x)) = e^{d(x)} \,/\, (MZ_0)$ (where $M$ is a constant with $M \geq p_d^*(x) \,/\, p_g(x)$), we get samples from the distribution $p_d^*$. We can view this rejection sampling as a rejection sampling in the latent space $\mathcal{Z}$, where we perform rejection sampling on $p_0(z)$ with acceptance probability $r(z) = p_d^*(G(z)) \,/\, (Mp_g(G(z))) = e^{d(G(z))} \,/\, M$. Applying lemma 1, we see that this rejection sampling procedure induces a probability distribution $p_t(z) = p_0(z)r(z)/C$ on the latent space $\mathcal{Z}$. $C$ is the normalization constant. Thus sampling from $p_d^*(x)$ is equivalent to sampling from $p_t(z)$ and generating with $G(z)$. $\qquad\square$

**Corollary 2.** *Assume $p_d$ is the data generating distribution with small Gaussian noise added. The generator $G : \mathcal{Z} \to \mathcal{X}$ is a deterministic function, where $\mathcal{Z}$ is the latent space endowed with prior distribution $p_0(z)$. Assume $z' \sim p_1(z') = N(0,1;z)$ is an additional Gaussian noise variable with $\dim z' = \dim \mathcal{X}$. Let $\epsilon > 0$, denote the distribution of the extended generator $G^*(z, z') = G(z) + \epsilon z'$ as $p_g$. $D(x)$ is the discriminator trained between $p_g$ and $p_d$. Let $d(x)$ be the logit of $D$, namely $D(x) = \sigma(d(x))$. Define $p_d^* = e^{\log p_g(x) + d(x)} \,/\, Z_0$, where $Z_0$ is the normalization constant. We define the energy function in the extended latent space $E(z, z') = -\log p_0(z) - \log p_1(z') - d(G^*(z, z'))$, and its Boltzmann distribution $p_t(z, z') = e^{-E(z,z')} \,/\, Z$. Then we have:*

1. *$p_d^* = p_d$ when $D$ is the optimal discriminator.*

2. *If we sample $(z, z') \sim p_t$, and $x = G^*(z, z')$, then we have $x \sim p_d^*$. Namely, the induced probability measure $G^* \circ p_t = p_d^*$.*

*Proof.* Let $G^*(z, z')$ be the generator $G$ defined in Theorem 1, we can see that $p_d$ and $p_g$ have the same support. Apply Theorem 1 and we deduce the corollary. $\qquad\square$

# B  An Analysis of WGAN

## B.1  An Analysis of the DOT algorithm

In this section, we first give an example that in WGAN, given the optimal discriminator $D$ and $p_g$, it is not possible to recover $p_d$.

Consider the following case: the underlying space is one dimensional space of real numbers $\mathbf{R}$. $p_g$ is the Dirac $\delta$-distribution $\delta_{-1}$ and data distribution $p_d$ is the Dirac $\delta$-distribution $\delta_a$, where $a > 0$ is a constant.

We can easily identity function $f(x) = x$ is the optimal 1-Lipschitz function which separates $p_g$ and $p_d$. Namely, we let $D(x) = x$ is the optimal discriminator.

However, $D$ is not a function of $a$. Namely, we cannot recover $p_d = \delta_a$ with information provided by $D$ and $p_g$. This is the main reason that collaborative sampling algorithms based on W-GAN formulation such as DOT could not provide exact theoretical guarantee, even if the discriminator is optimal.

## B.2  Mathematical Details of Approximating WGAN with EBMs

In the paper, we show that the optimization of WGAN can be viewed as an approximation of an energy-based model. We present more details here.

For Eq. (3):

$$
\begin{aligned}
\nabla_\phi \mathrm{KL}(p_d||p_t) =& \nabla_\phi \mathbb{E}_{p_d}[-\log p_t(x)] \\
=& \nabla_\phi \mathbb{E}_{p_d}[-\log p_g(x) - D(x) + \log Z] \\
=& -\mathbb{E}_{p_d}[\nabla_\phi D(x)] + \mathbb{E}_{p_d}[\nabla_\phi \log Z] \\
=& -\mathbb{E}_{p_d}[\nabla_\phi D(x)] + \nabla_\phi Z/Z \\
=& -\mathbb{E}_{p_d}[\nabla_\phi D(x)] + \sum_x [p_g(x)e^{D(x)}\nabla_\phi D(x)]/Z \\
=& -\mathbb{E}_{p_d}[\nabla_\phi D(x)] + \sum_x [p_t(x)\nabla_\phi D(x)] \\
=& \mathbb{E}_{p_t}[\nabla_\phi D(x)] - \mathbb{E}_{p_d}[\nabla_\phi D(x)]
\end{aligned}
\tag{6}
$$

For Eq. (4):

$$
\begin{aligned}
\nabla_\theta \mathrm{KL}(p_g||p'_t) =& \nabla_\theta \mathbb{E}_{p_g}[\log p_g(x) - \log p'_t(x)] \\
=& \mathbb{E}_{p_g}[\nabla_\theta \log p_g(x)] + \sum_x [\log p_g(x) - \log p'_t(x)]\nabla_\theta p_g(x) \\
=& 0 + \sum_x [-D(x)]\nabla_\theta p_g(x) \\
=& -\sum_x D(x)\nabla_\theta p_g(x) \\
=& -\nabla_\theta \mathbb{E}_{p_g}[D(x)] = -\mathbb{E}_{z \sim p_0(z)}[\nabla_\theta D(G(z))]
\end{aligned}
\tag{7}
$$

# C  Experimental details

Source code of all experiments of this work is included in the supplemental material , where all detailed hyper-parameters can be found.

## C.1  Synthetic

The 25-Gaussians dataset is generated by a mixture of twenty-five two-dimensional isotropic Gaussian distributions with variance 0.01, and means separated by 1, arranged in a grid. The Swiss Roll dataset is a standard dataset for testing dimensionality reduction algorithms. We use the implementation from

scikit-learn, and rescale the coordinates as suggested by [12]. We train a Wasserstein GAN model with the standard WGAN-GP objective. Both the generator and discriminator are fully connected neural networks with ReLU nonlinearities, and we follow the same architecture design as in DOT [12], while parameterizing the prior with a standard normal distribution instead of a uniform distribution. We optimize the model using the Adam optimizer, with $\alpha = 0.0001, \beta_1 = 0.5, \beta_2 = 0.9$.

## C.2 CIFAR-10 and CelebA

Figure 3: CIFAR-10 Langevin dynamics visualization, initialized at a sample from the generator (left column). The latent space Markov chain appears to mix quickly, as evidenced by the diverse samples generated by a short chain. Additionally, the visual quality of the samples improves over the course of sampling, providing evidence that DDLS improves sample quality.

Figure 4: Top-5 nearest neighbor images (right columns) of generated CIFAR-10 samples (left column).

For CIFAR-10 dataset, we adopt the Spectral Normalization GAN (SN-GAN) [1] as our baseline GAN model. We take the publicly available pre-trained models of unconditional SN-GAN and apply DDLS. For CelebA dataset, we adopt DCGAN and WGAN as the baseline model following the same setting in [56]. We first sample latent codes from the prior distribution, then run the Langevin dynamics procedure with an initial step size 0.01 up to 1000 iterations to generate enhanced samples. Following the practice in [57] we separately set the standard deviation of the Gaussian noise as 0.1. We optionally fine-tune the pre-trained discriminator with an additional fully-connected layer and a logistic output layer using the binary cross-entropy loss to calibrate the discriminator as suggested by [10, 11].

We show more generated samples of DDLS during langevin dynamics in Fig. 6. We run 1000 steps of Langevin dynamics and plot generated sample for every 10 iterations. *We include* 10000 *more randomly generated samples in the supplemental material.*

## C.3 Imagenet

We introduce more details of the preliminary experimental results on Imagenet dataset here. We run the Langevin dynamics sampling algorithm with an initial step size 0.01 up to 1000 iterations. We decay the step size with a factor 0.1 for every 200 iterations. The standard deviation of Gaussian noise is annealed simultaneously with the step size. The discriminator is not yet calibrated in this preliminary experiment.

Figure 5: Progression of Inception Score with more Langevin dynamics sampling steps.

---

**Algorithm 1** Discriminator Langevin Sampling

---

**Input:** $N \in \mathbb{N}_+, \epsilon > 0$
**Output:** Latent code $z_N \sim p_t(z)$
Sample $z_0 \sim p_0(z)$.
**for** $i < N$ **do**
    $n_i \sim N(0,1)$
    $z_{i+1} = z_i - \epsilon/2 \nabla_z E(z) + \sqrt{\epsilon} n_i$
    $i = i + 1$
**end for**

---

## D  DDLS Algorithm

We show the detailed algorithm using Langevin dynamics in Alg. 1.

## E  Hybrid WGAN-EBM Training Algorithm

In Sec. 3.4, we described an EBM algorithm which WGAN is approximately optimizing. Here we detail this algorithm in Alg. 2.

---

**Algorithm 2** WGAN-EBM Hybrid Algorithm

---

**Input:** $N \in \mathbb{N}_+, \epsilon > 0, \delta > 0$, Initialized $D_\phi(x), G_\theta(z)$
**Output:** Trained $D_\phi(x), G_\theta(z)$
**for** Model Not Converged **do**
    Sample a batch $z_0^k \sim p_0(z), k = 1, 2, \cdots M$.
    Sample a batch of real data $x^k, k = 1, 2, \cdots M$.
    **for** $i < N$ **do**
        $n_i^k \sim N(0,1)$
        $z_{i+1}^k = z_i^k - \epsilon/2 \nabla_z E(z_i^k) + \sqrt{\epsilon} n_i^k, \{E(z) = -\log p_0(x) - D(G(z))\}$
        $i = i + 1$
    **end for**
    $\phi = \phi - \delta(\sum_{k=1}^M \nabla_\phi D(G(z_N^k)) - \nabla_\phi D(x^k))$
    $\theta = \theta + \delta \sum_{k=1}^M \nabla_\theta D(G(z_0^k))$
**end for**

---

Figure 6: CIFAR-10 Langevin dynamics visualization