[Reviews · NeurIPS 2020]

Review 1

Summary and Contributions: The paper introduces a method to perform sampling in the latent space of GANs by formulating an energy-based model in the latent space. Employing rejection sampling, the authors devise a tractable energy function in the latent space that utilizes a discriminator score. Langevin dynamics can then be used to sample from the induced Boltzmann distribution. === Post rebuttal and discussion update === I acknowledge the rebuttal and discussion, I believe that the confusion with the MCMC mixing pointed out by the reviews should definitely be addressed in the revision and the FID scores should be computed for at least CelebA, but I still tend to keep my score.

Strengths: This work aims at improving the sample quality of generative models through better sampling, which is a relevant problem and has brought about a line of work, [1,2,3,4], to name a few. This paper treats the problem by an energy-guided search for samples in the latent space rather than operating in the pixel space, which proves to be favourable empirically - both quantitatively and qualitatively. It also builds on a recent interest to energy-based model interpretation for GANs allowing theoretical guarantees***, analysis and extension to the WGAN case.*** [1] Azadi, Samaneh, et al. "Discriminator rejection sampling." arXiv preprint arXiv:1810.06758 (2018). [2] Turner, Ryan, et al. "Metropolis-hastings generative adversarial networks." International Conference on Machine Learning. 2019. [3] Neklyudov, Kirill, Evgenii Egorov, and Dmitry P. Vetrov. "The Implicit Metropolis-Hastings Algorithm." Advances in Neural Information Processing Systems. 2019. [4] Tanaka, Akinori. "Discriminator optimal transport." Advances in Neural Information Processing Systems. 2019.

Weaknesses: The paper (lines 113-114) creates an impression that the method doesn't require a generator pass to sample a better latent code. However, to compute the gradient of the energy, the algorithm actually needs a discriminator score (generator + discriminator pass) meaning that each iteration of Langevin dynamics requires a generated sample from updated z (increasing computational complexity to arrive at a better sample in pixel space). This should be made more explicit in the paper if my understanding is correct.

Correctness: Claims and method seem good, empirical methodology is in line with the previous work.

Clarity: Sections 3.2 and 3.3 would benefit from a bit more comprehensible exposition. Also, some references to Figures (3+) and Table 4 located in Appendix are confusing since it is not stated where to find them. Apart from that, the paper is well written.

Relation to Prior Work: Yes, the related work section discusses most of the relevant work and gives a clear distinction. However, I suggest clarifying the fact that there is a previous work that also carries out MCMC sampling in the latent space, e.g. [5]. [5] Kumar, Rithesh, et al. "Maximum entropy generators for energy-based models." arXiv preprint arXiv:1901.08508 (2019).

Reproducibility: Yes

Additional Feedback: It would be great to have a succinct comparison with a concurrent paper [6] in the related work. [6] Arbel, Michael, Liang Zhou, and Arthur Gretton. "KALE: When Energy-Based Learning Meets Adversarial Training." arXiv preprint arXiv:2003.05033 (2020). (as far as I know, it's now reworked into Generalized Energy Based Models - [https://arxiv.org/abs/2003.05033](https://arxiv.org/abs/2003.05033)) There are also a few misprints / minor comments: 1. I believe Z in lines 134 and 139 is the same but has an additional prime mark when it's usedin line 139. 2. line 190 following 3. line 300 says that fine tuning is described in 5.2, but it's missing there. (it's in Appendix) 4. Supplementary material has wrong enumeration for Lemma, Theorem and Corollary.


Review 2

Summary and Contributions: This paper proposed Discriminator Driven Latent Sampling(DDLS), which aims to improve the generation quality by adding an extra “selection” process based on MCMC In the latent space. The method draws a connection between the GANs and energy-based models. Empirical results on both the synthetic setting and real-world dataset have demonstrated the improvement of the prosed method on standard evaluation metrics such as inception score. POST REBUTTAL: I appreciate the authors' efforts to involve further experiments. As the concern on the experiment and evaluation remains, I tend to keep my score.

Strengths: It is interesting to draw connections between GANs and energy-based models. It's reasonable that the energy function which is jointly implied by the discriminator and generator may define a superior distribution than the generated distribution. The theoretical discussion is well organized and easy to understand. The overall writing of the paper is sound. Experiment results on cifar-10 and the synthetic setting are impressive.

Weaknesses: One major concern is that the novelty of the paper is limited. In the f-divergence settings, the key motivation is to leverage the density ratio estimation property of the discriminator to construct a new distribution which is assumed to be superior. As the author mentioned, it has been widely explored by previous work such as DRS and MHGAN. Especially, the only difference between the DDLS and MHGAN lies in the "filtering" happens in sample space or latent space. Essentially,MHGAN could be understood as using an independent proposal in the latent space and DDLS may enjoy the good property of HMC. However, the stationary distribution remains the same and such modification seems to be marginal. In the WGANs' setting, the connection between the GANs and EBM is derived by Eq. 3 which needs an assumption that pt and pg are close enough as the author claimed. However, such an assumption is hard to satisfy, i.e. more regularization on the generator is needed as shown in [1,2], which may explain the difficulties on sampling from pt in the pixel space. Hence, with such a strong assumption, the derived connection seems to be unnatural. The evaluation of the proposed method is not solid enough. The FID score is only evaluated on CIFAR-10. How is the FID on celeba and imagenet? Whether the proposed method overfit on the inception score? As diversity preservation is considered an important advantage of the proposed method, it is important to report the FID number which is recognized as a better evaluation metric on diversity than the inception score. [1] Maximum Entropy Generators for Energy-Based Models [2] Exponential Family Estimation via Adversarial Dynamics Embedding

Correctness: yes

Clarity: yes

Relation to Prior Work: not sure.

Reproducibility: Yes

Additional Feedback: Introducing an additional noise to solve the mode dropping problem is interesting, is it possible to conduct an ablation study to test the effectiveness on the real world datasets?


Review 3

Summary and Contributions: This paper proposes DDLS that runs MCMC in the latent space and shows superiority compared with vanilla GAN and other sampling assisted GANs.

Strengths: The new sampling based GAN model performs better than other related works and may inspires some more research working on latent MCMC sampling.

Weaknesses: ------------Post Rebuttal----------------------- I appreciate the authors' responses. However, these don't address my concern very well. The paper claims that the model has a fast mixing. I don't see it in the paper. A good mixing means the a single MCMC can approximate the target distribution and traverse almost all modes. Your model just corrects the samples from GAN. This paper has a very serious misunderstanding about the definition of MCMC mixing and mistakenly uses Fig.5 as a support. Fig.5 only shows a result from independent MCMC. The proposed method doesn't provide a guarantee for the MCMC convergence. I think this is very misleading for future research. Please refer to https://arxiv.org/pdf/1207.4404.pdf. I will keep my original score, a clear rejection. But I encourage the authors to revise the paper and make the contribution clear. 1. Motivation is weak. In line 25 - 26, the author claims that bad artifacts exist in high-resolution images, or are even not recognizable. However, the proposed model only applies to very low resolution cifar dataset and doesn't explicitly address this issue. Please refer to MH-GAN. 2. In liine 34, why do you claim that your method is more efficient than MH-GAN? Rejection sampling are used in both cases. Or does MH-GAN lack theoretical guarantee? Maybe you can show your method leads to a fast mixing time compared with related works. 2. Where are the generated samples that can show diversity and fidelity, e.g. imagenet and celeba? Quantitative results which only shows inception score in Table 4 are not convincing. What about the FID? 3. In line 100, the author claims MCMC in pixel space it not applicable. However, a lot of works have been working successfully even on ImageNet128, like [1][2] and so on... [1] Implicit generation and modeling with energy-based models, in Neurips 2019 [2] A theory of generative model, in ICML 2016

Correctness: Yes.

Clarity: Yes.

Relation to Prior Work: Not accurate.

Reproducibility: Yes

Additional Feedback:


Review 4

Summary and Contributions: The paper proposes a new view to analysis GAN from the latent space energy based model, and demonstrates that by doing Langevin on the latent space EBM instead of high-dimensional pixel space EBM, the generator sample quality can be improved.

Strengths: The idea presented in the paper is well motivated and clear. The connection between latent space EBM and GAN is insightful. The work should be of broad interest to the generative modelling as well as unsupervised learning fields.

Weaknesses: No major flaws. The paper could be better if: (1) add more motivations on section 3.4 (wgan with langevin), e.g., what is the main motivation to consider p_t = p_g exp^{D}/Z (line 156), it has similar form as the one derived from DCGAN setting (based on the optimality of the generator/discriminator), how about in the WGAN setting here? To make sample sharper? what else? (2) Some figures (e.g., fig 4 & 5 etc) are referred in the main text, perhaps should state clearly in the revision that such figures are in appendix.

Correctness: The model seems correct

Clarity: The paper is well written

Relation to Prior Work: It clearly discussed the difference from the previous arts.

Reproducibility: Yes

Additional Feedback: The author rebuttal addressed some of my concerns, I still feel its a nice work and could be of broad interest to NeurIPS community.

[Author Response · NeurIPS 2020]

We thank all the reviewers for the positive feedback and thoughtful comments. Below we address all the comments and questions in order.

On the concern of reviewer 2 and reviewer 3 about the novelty and efficiency of DDLS:

**0**. We ran additional experiments to include the FID score progression during the MH/Langevin dynamics, with direct comparasion to MH-GAN baseline. We adopt the same WGAN architecture in our own implementation and report FID score for each 20 iterations on CIFAR-10, up to 640 iterations following the same setting in MH-GAN. We see that DDLS in latent space enjoys much faster mixing in the first 200 iterations.

Figure 1: Progression of FID Score with MH/Langevin dynamics sampling steps, averaged by multiple runs.

**1**. Although the stationary distribution which we are sampling from in the vanilla GAN case is the same as DRS/MH-GAN, this stationary distribution was intractable before our work. Both DRS and MH-GAN had to use a proposal rejection/acceptance scheme. Our work proves that the distribution was indeed tractable in the latent space, with a clear and insightful energy-based model (EBM) formulation. This is our main technical contribution. The Langevin sampling scheme is only made possible thanks to tractable EBM formulation. So we think it's unfair to say that our contribution is just to replace MH sampling in MH-GAN with Langevin dynamics. We also extended the formulation to other GANs such as WGANs and shows its efficiency in our experiments.

**2**. In theory, DRS and MH-GAN samples from the same stationary distribution as ours, however in practice, independent sampling schemes such as DRS or MH-GAN can be very inefficient, and the number of steps required to move the sampler distribution to the stationary distribution can be too large for any reasonable computing resources. For example, consider training a generator on the MNIST dataset, where the generator produces 0.1% of number "0", when real data contains 10% of number "0" (this is common for GANs which are not very good at balancing different modes). Then if we use MHGAN to generate 100 numbers to simulate the real data distribution, in which 10 numbers should be "0", you have to generate 10,000 and reject 9,000 of them, even if these samples are perceptually good. In realistic cases where only a finite number of samples can be used, this inefficiency will seriously hurt the resulting distribution sampled by MHGAN. However, with our method, the gradient of the discriminator can guide the Langevin dynamics to move towards 0, which can be much more efficient. As shown in Figure. 1, we compare the FID of different sampling steps of the three methods on CIFAR-10. The FID score of our method goes down quickly with just a few MCMC steps, while for MH-GAN, the score goes down slowly, and stabilized at a much higher value. This observation confirms our claim that although in theory MH-GAN can achieve the same distribution of our method, it may take unbounded sampling steps to achieve that. So in practice the sampling distribution is much worse than in our work.

Other concerns:

Reviewer 1: Thanks for the valuable comments! Our method does need a generator pass to get better $z$. We will improve the writing of the paper to make it clearer. We will also include more discussions about the related works mentioned.

Reviewer 2: On the WGAN assumption, $p_t$ and $p_g$ can be close if $p_g$ is close enough to $p_d$, this can be achieved after the training. Note our definition of $p_t = p_g(x)e^{-E(x)}/Z$ is a product distribution which is different from $p_t(x) = e^{-E(x)}/Z$ in [1,2]. In their definition, $p_t$ and $p_g$ may not be very close. Also, in LOGAN, the training explicitly takes gradient steps in latent space, which makes their distribution of negative samples more close to $p_t$. Please also refer to R.3 below for the FID score issue.

Reviewer3: Images with large resolution requires extra computational resources which we didn't have at the moment of submission, but we will add them in the next version. We have discussed why our method is much more efficient than MH-GAN in detail above. We didn't include the FID score on CelebA since the lack of FID scores in baseline methods, which makes direct comparaions impossible.

We didn't claim doing MCMC in the pixel space is not possible at all, but it is inefficient and hard to tune. Pixel-level EBMs are very sensitive to hyperparameters, needs a bunch of training tricks, very slow to train, and their performance is not comparable with GANs and our model.

Reviewer4: The main motivation of considering $p_t = p_g(x)e^{D(x)}/Z$ is to find an approximate energy-based model which is close enough to the actual WGAN model. In this way we can do latent space MCMC and get a distribution which is much closer to the data distribution than the original generator distribution. We will revise the paper to make it more clear, thanks for your comment.

[Meta-Review · NeurIPS 2020]

Two reviewers strongly recommend acceptance, while two recommend reject. R1 and R4 argue that the proposed connection between energy networks and GANs is new and will be of general interest to the community, and the results on CIFAR-10 are sufficient to demonstrate the utility of the proposed approach. R2 and R3 argue that the results are unconvincing because important evaluations are missing (FID score) on CelebA. I take this point, and encourage authors to add these results. Further, R3 points out that Figure 5 doesn't actually demonstrate MCMC mixing as claimed. However, overall I agree with R1 and R4 that the positives outweigh the negatives. I recommend acceptance, but strongly encourage authors to take reviewer suggestions seriously and implement changes for camera ready. The wording around MCMC mixing and Figure 5 should be clarified in camera ready to avoid confusion.